# TIGR-Tas and the Expanding Universe of RNA-Guided Genome Editing Systems: A New Era Beyond CRISPR-Cas

**DOI:** 10.3390/genes16080896

**Published:** 2025-07-28

**Authors:** Douglas M. Ruden

**Affiliations:** Department of Obstetrics and Gynecology, C. S. Mott Center for Human Growth and Development, Institute of Environmental Health Sciences, Wayne State University, Detroit, MI 48201, USA; douglasr@wayne.edu; Tel.: +1-313-577-6688

**Keywords:** TIGR-Tas, RNA-guided genome editing, CRISPR-Cas systems, dual-spacer tigRNA, programmable DNA nucleases, PAM-independent targeting, Tas proteins, synthetic biology, genome engineering

## Abstract

The recent discovery of TIGR-Tas (Tandem Interspaced Guide RNA-Targeting Systems) marks a major advance in the field of genome editing, introducing a new class of compact, programmable DNA-targeting systems that function independently of traditional CRISPR-Cas pathways. TIGR-Tas effectors use a novel dual-spacer guide RNA (tigRNA) to recognize both strands of target DNA without requiring a protospacer adjacent motif (PAM). These Tas proteins introduce double-stranded DNA cuts with characteristic 8-nucleotide 3′ overhangs and are significantly smaller than Cas9, offering delivery advantages for in vivo editing. Structural analyses reveal homology to box C/D snoRNP proteins, suggesting a previously unrecognized evolutionary lineage of RNA-guided nucleases. This review positions TIGR-Tas at the forefront of a new wave of RNA-programmable genome-editing technologies. In parallel, I provide comparative insight into the diverse and increasingly modular CRISPR-Cas systems, including Cas9, Cas12, Cas13, and emerging effectors like Cas3, Cas10, CasΦ, and Cas14. While the CRISPR-Cas universe has revolutionized molecular biology, TIGR-Tas systems open a complementary and potentially more versatile path for programmable genome manipulation. I discuss mechanistic distinctions, evolutionary implications, and potential applications in human cells, synthetic biology, and therapeutic genome engineering.

## 1. Introduction

Programmable genome editing has undergone a dramatic transformation since the discovery and repurposing of the CRISPR-Cas9 system. This bacterial immune mechanism, centered on Cas proteins guided by RNA to cleave nucleic acid targets, has become the dominant toolset in modern biotechnology [1]. Yet, as researchers delve deeper into microbial genomes and viral metagenomes, new families of RNA-guided systems are emerging—some with profound functional divergence from canonical Cas effectors (Table 1).

Among these, TIGR-Tas (Tandem Interspaced Guide RNA-Targeting Systems) represents a paradigm shift. Recently described in both parasitic bacteria and their associated phages, TIGR-Tas systems feature compact Tas proteins that are guided by tigRNAs, which is a novel class of dual-spacer RNAs that bind both strands of a DNA target in the absence of a PAM sequence (Graphic Abstract). Upon binding, the Tas effector induces a double-stranded break with a clean and defined 8-nucleotide 3′ overhang. This dual-strand binding mechanism of tigRNAs stands in contrast to the single-strand binding by crRNAs (CRISPR RNAs) of the CRISPR-Cas family, which is thought to obviate the need for the PAM sequence and dramatically increase the target range of cleavage sites [2].

The small size of Tas proteins—approximately one quarter the molecular weight of Cas9—combined with their PAM independence and ability to generate defined 8-nucleotide 3′ overhangs—positions TIGR-Tas as a highly promising platform for compact and precise genome editing [2]. These features make TIGR-Tas especially attractive for applications in therapeutic delivery, crop engineering, and synthetic biology. Unlike canonical CRISPR effectors, Tas proteins share minimal sequence homology with known Cas enzymes but exhibit distinctive structural elements, including the Nop (nucleolar protein) domain, which is conserved across several pre-RNA processing ribonucleoproteins (RNPs) [3]. This domain is notably found in eukaryotic box C/D small nucleolar RNPs (snoRNPs), which mediate the site-specific 2′-O-ribose methylation of ribosomal RNA [4]. The presence of a Nop domain in Tas proteins suggests an unexpected evolutionary link between prokaryotic DNA-targeting systems and the eukaryotic RNA-guided complexes. This observation raises the possibility that sequence-specific RNA-guided nucleoprotein systems have evolved multiple times independently across domains of life.

While CRISPR-Cas effectors remain foundational tools in molecular biology, continued innovation has broadened their capabilities far beyond simple double-strand DNA cleavage. Cas9, the best-known and most widely used effector, generates blunt double-stranded breaks (DSBs) guided by a single-guide RNA (sgRNA) but has also been engineered into nickase variants (nCas9) that cleave only one strand of DNA, enabling more precise applications such as base editing and prime editing [5]. Furthermore, catalytically inactive Cas9 (dCas9) has been fused to a variety of effector domains [6], including transcriptional activators (e.g., VP64, p300) [7] and repressors (e.g., KRAB) [8], enabling programmable gene regulation without altering the DNA sequence.

Cas12 offers an alternative, generating staggered DSBs and supporting multiplexed editing through its ability to process multiple crRNAs from a single transcript [9,10]. Cas13, by contrast, exclusively targets RNA, providing a powerful tool for transcriptome modulation, viral RNA degradation, and sensitive diagnostics through collateral cleavage activity [11,12]. Other Cas effectors, such as Cas3, which exhibits processive DNA degradation (DNA shredding) [8], and Cas10, which participates in dual RNA-DNA interference via cyclic oligoadenylate signaling [13,14], contribute to specialized functions across CRISPR system subtypes (Table 1).

Despite this extensive functional repertoire, TIGR-Tas introduces fundamentally new capabilities. Unlike Cas proteins, Tas effectors require no PAM for DNA targeting, expanding the editable genome space. Their use of dual-spacer tigRNAs to simultaneously recognize both strands of the DNA duplex allows precise targeting and cleavage that results in 8-nucleotide 3′ overhangs, which is a feature not observed in any known Cas system. Additionally, Tas proteins are much smaller—roughly one quarter the size of Cas9—facilitating delivery in size-constrained vectors such as AAVs. TIGR-Tas systems, therefore, represent not only an evolutionary departure from CRISPR-Cas mechanisms but also a functional expansion of RNA-guided editing modalities [2].

This review highlights the molecular architecture, guide RNA structure, and cleavage mechanics of TIGR-Tas and compares its unique advantages with the expanding CRISPR-Cas toolkit (Graphic Abstract). By contextualizing TIGR-Tas within this broader landscape, I aim to underscore its transformative potential for compact, precise, and PAM-independent genome editing. I aim to catalyze the further study and application of these RNA-guided, PAM-less, programmable nucleases, which may soon join—or even rival—the most widely used tools in genome science.

## 2. Materials and Methods

I conducted a comprehensive review of the literature on RNA-guided nucleases, focusing on the recently published characterization of TIGR-Tas systems [2]. Databases searched included PubMed, NCBI, and CRISPRCasdb. Search terms included “TIGR-Tas”, “Tas protein”, “tigRNA”, “RNA-guided nucleases”, and specific Cas proteins (e.g., “Cas9”, “Cas12”, “Cas13”, “Cas3”, “Cas10”, “CasΦ”, “Cas14”) (Table 1).

Primary research articles, structural studies, and metagenomic surveys were prioritized, particularly those that defined PAM requirements, cleavage mechanics, guide RNA architectures, and evolutionary relationships. Data on TIGR-Tas cleavage behavior, guide structure, and in vivo activity in human cells were extracted from the primary publication [2]. Comparative data for Cas effectors were taken from published reviews and mechanistic studies from 2014 to 2025.

To ensure breadth and currency, the review includes recent discoveries such as CasΦ and Cas14 as well as the long-established Cas9/Cas12/Cas13 enzymes, which provide the functional backdrop against which TIGR-Tas systems are now emerging.

**Table 1 genes-16-00896-t001:** Overview of CRISPR and TIGR-Tas systems.

System	MW (kDa) ^1^	Short Description	Reference(s)
Cas9	162	Cuts double-stranded DNA (dsDNA) guided by single guide RNA (sgRNA)	[15]
Cas12a	156	Staggered dsDNA cut guided by CRISPR RNA (crRNA); recognizes T-rich PAM; collateral RNA activity	[9,10]
Cas13a	144	Single-stranded RNA (ssRNA) cleavage via crRNA; collateral RNA activity; can be used for the rapid detection of RNA viruses	[11,12]
Cas14a	40	Ultra-small protein that cuts single-stranded DNA (ssDNA); target recognition by Cas14 triggers nonspecific ssDNA cleavage, enabling high-fidelity SNP genotyping (Cas14-DETECTR)	[16]
Cas3	100	Helicase-nuclease that shreds ssDNA unidirectionally after Cascade complex binding;	[17,18]
Cas10	80	Targets both DNA and RNA via crRNA; part of Type III systems; uses cyclic oligoadenylates (cOA) as second messengers to activate the Csm6 nuclease to promote RNA degradation	[13,14]
Cas1–Cas2	78 ^2^	Integrates foreign DNA spacers into CRISPR array; core adaptation machinery	[19,20]
CasΦ (CasPhi)	70	Hypercompact dsDNA-cutting protein from giant phages	[21]
CasΨ (CasPsi)	90	A highly specific nuclease sensitive to SNPs next to the PAM; a.k.a., Cas12j	[22,23]
TIGR-Tas	36	PAM-less dsDNA cleavage by Tas proteins guided by dual-spacer tigRNA; creates 8-nt 3′ overhangs	[2]

^1^ The approximate MW (molecular weight) of the smallest protein in this family. ^2^ The MW of the smallest known tetramer.

## 3. Discussion

### 3.1. The Two Main Classes of CRISPR-Cas Proteins

CRISPR-Cas systems are classified into two main classes based on how they work to defend against invaders like viruses.

Class 1 systems use multiple Cas proteins that form a complex to find and interfere with target DNA or RNA. These systems are common in nature; they are found in most bacteria and nearly all archaea. Class 1 includes types I, III, and IV [24,25]. Type I systems use Cas3 to cut and degrade DNA [26]. Type III systems can cut both DNA and RNA, using proteins such as Cas10, and they have a mechanism to avoid attacking the cell’s own DNA that involves novel cyclic oligoadenylates (cOA) as second messengers [27]. Type IV systems involve a smaller multi-protein complex [28,29].

Class 2 systems use a single, large protein to perform the interference. These systems are less common in nature than Class 1, but they are known for their use in gene editing [30,31]. Class 2 includes types II, V, and VI. Type II systems, such as Cas9, also require a guide RNA called tracrRNA [32,33]. Type V systems, including those that use Cas12, also target DNA [9,10]. Type VI systems, using Cas13, specifically target collateral RNA [11,12]. The main difference is that Class 1 uses a multi-protein effector complex, while Class 2 uses a single effector protein, which affects their function and the molecules they target.

Some of the more common CRISPR-Cas family members are described in the next subsections.

#### 3.1.1. Cas9: The Most Widely Used Tool for Genome Editing and High-Throughput Screens

The CRISPR-Cas9 system remains the most widely utilized and versatile tool in the RNA-guided genome editing landscape. Cas9, the hallmark effector of Type II CRISPR systems, relies on the recognition of a short DNA sequence known as the protospacer adjacent motif (PAM)—typically NGG for *Streptococcus pyogenes* Cas9—to distinguish foreign targets from self-derived CRISPR arrays. Target recognition begins when Cas9, loaded with a guide RNA, scans the genome for PAM sequences. Upon PAM detection, the adjacent DNA is interrogated by the complementary spacer sequence within the guide RNA. If sufficient base pairing occurs, Cas9 introduces a blunt double-stranded break at the target site via its RuvC (repairs ultraviolet damage), originally discovered as binding to DNA Holliday structures [34], and HNH nuclease domains (His-Asn-His), which cleave the non-target and target strands, respectively [15].

In engineered systems, the traditional two-RNA structure—consisting of a CRISPR RNA (crRNA) and a trans-activating crRNA (tracrRNA)—has been fused into a single guide RNA (sgRNA), streamlining the system for laboratory use [35,36]. The crRNA component confers sequence specificity, while the tracrRNA scaffold enables Cas9 binding and activation.

One of the most impactful applications of CRISPR-Cas9 has been the development of high-throughput functional genomic screens. These platforms use libraries of thousands of sgRNAs targeting nearly every gene in the genome to systematically perturb gene function in pooled populations of cells [37]. By coupling sgRNA barcodes with next-generation sequencing, researchers can identify essential genes, synthetic lethal interactions, drug-resistance mechanisms, and regulators of specific phenotypes. Screens can be tailored for knockout, activation, or repression modalities using Cas9, dCas9, or Cas9-based fusions, and they have transformed both basic biology and drug discovery pipelines [37].

Collectively, the adaptability of Cas9 through rational engineering and synthetic fusion has solidified its central role in both basic research and therapeutic development.

#### 3.1.2. Cas12: A Versatile Type V DNA Editor and Diagnostic Tool

Cas12 (also known as Cpf1) belongs to Class 2, type V CRISPR-Cas systems and offers several key mechanistic and functional differences from Cas9. While Cas9 recognizes G-rich PAM sequences (typically NGG), Cas12 effectors preferentially target T-rich PAMs (commonly 5′-TTTV-3′, where V = A/C/G), significantly expanding the range of accessible genomic sites for editing. Upon binding to a target, Cas12 introduces staggered double-stranded DNA breaks, generating 5′ overhangs—contrasting with Cas9’s blunt-ended cuts. These staggered breaks can enhance homology-directed repair (HDR) efficiency and facilitate precise insertions or knock-ins [16].

A major advantage of Cas12 lies in its simplified guide RNA architecture. Unlike Cas9, which requires both a crRNA and a tracrRNA (or a synthetic sgRNA), Cas12 operates using only a single crRNA, making it more amenable to compact and programmable designs for synthetic biology or therapeutic delivery platforms.

One of Cas12’s most distinctive features is its collateral single-stranded DNA (ssDNA) cleavage activity. Upon target recognition and activation, Cas12a begins the indiscriminate cleavage of nearby ssDNA molecules. While initially considered a challenge for genome editing fidelity, this property has been repurposed to develop ultrasensitive molecular diagnostics, including the DETECTR platform (DNA Endonuclease-Targeted CRISPR Trans Reporter). DETECTR couples Cas12’s collateral activity to fluorescent or colorimetric readouts, enabling the rapid, point-of-care detection of viral, bacterial, and human nucleic acid targets, including SARS-CoV-2, HPV, and cancer mutations [16].

Beyond diagnostics, Cas12 effectors support multiplex genome editing, owing to their ability to process precursor CRISPR arrays into multiple mature crRNAs without requiring additional proteins. This autonomous processing allows the simultaneous targeting of multiple genomic loci from a single transcript, streamlining design for complex genetic modifications in plants, animals, and microbes [38].

Variants such as Cas12b (C2c1) and Cas12f (Cas14) offer further functional diversity, including smaller protein size, expanded PAM recognition profiles, and potential for future engineering into base editors or prime editors. With its unique combination of compact architecture, programmable targeting, collateral detection capability, and multiplexing potential, Cas12 has emerged as a powerful tool not only for genome editing but also for biosensing, gene regulation, and point-of-care molecular diagnostics [37].

#### 3.1.3. Cas13: A Programmable RNA-Targeting Tool for Editing, Detection, and Regulation

Cas13 is the signature effector of Class 2, type VI CRISPR-Cas systems and stands out for its unique ability to target and cleave RNA rather than DNA. This RNA-centric activity positions Cas13 as a transformative tool for transcriptome engineering, post-transcriptional regulation, and molecular diagnostics [37].

Cas13 effectors are guided by a single CRISPR RNA (crRNA) that contains a direct repeat and a variable spacer sequence complementary to the target single-stranded RNA (ssRNA). Upon successful recognition and binding to the target RNA, Cas13 undergoes a conformational change that activates its HEPN (Higher Eukaryotes and Prokaryotes Nucleotide-binding) domains—dual catalytic motifs responsible for RNA cleavage. Cas13 exhibits both cis-cleavage of the intended target and trans-cleavage (collateral activity) in which surrounding non-target ssRNAs are also degraded [37].

This collateral cleavage behavior, once viewed as a limitation, has been ingeniously adapted for molecular diagnostics. The SHERLOCK (Specific High-sensitivity Enzymatic Reporter unLOCKing) platform harnesses Cas13’s trans-cleavage to detect RNA with attomolar sensitivity. In SHERLOCK, target RNA binding activates Cas13, which subsequently cleaves a quenched reporter RNA, generating a detectable fluorescence or colorimetric signal. This has enabled rapid, field-deployable assays for viral pathogens (e.g., Zika, Ebola, SARS-CoV-2), cancer mutations, and even SNP genotyping [39,40].

Beyond diagnostics, Cas13 has emerged as a potent tool for programmable RNA knockdown, providing an alternative to RNAi with enhanced specificity and fewer off-target effects. Cas13-mediated knockdown has been used in human cells, plants, and model organisms for functional genomics and therapeutic applications. Moreover, engineered Cas13 variants—including catalytically dead versions (dCas13)—have been fused to effector domains for transcript localization, RNA editing, and splicing modulation [41]. For example, dCas13 fused to ADAR deaminases can enable A-to-I RNA editing, enabling the precise, transient modification of RNA without altering the genome [42,43].

The family of Cas13 enzymes is diverse, including subtypes such as Cas13a, Cas13b, Cas13c, and Cas13d, each with distinct sizes, target preferences, and processing mechanisms. Cas13d is notable for its compact size (~930 amino acids), making it suitable for AAV (Adeno-associated virus)-mediated delivery and in vivo applications [43,44].

In summary, Cas13 extends the CRISPR toolkit from static genome manipulation to dynamic transcriptome control, enabling diagnostic, therapeutic, and basic science applications that require precise, programmable RNA targeting.

#### 3.1.4. Cas3: A DNA Shredding Enzyme for Large-Scale Genome Remodeling

Cas3 is the signature nuclease-helicase of Type I CRISPR-Cas systems, which are the most abundant and evolutionarily ancient CRISPR systems in nature. Unlike Class 2 effectors like Cas9 and Cas12, which operate as single multidomain proteins, Cas3 functions in a Class 1 multi-protein complex known as Cascade (CRISPR-associated complex for antiviral defense). The Cascade complex, composed of multiple Cas subunits and a guide crRNA, scans the genome for a complementary protospacer adjacent to a PAM sequence (typically 5′-AAG or other variations depending on subtype) [43,44].

Once the Cascade-crRNA complex binds the target DNA through Watson-Crick base pairing, it undergoes a conformational change that exposes the adjacent DNA strand and recruits Cas3. Cas3 is a dual-function enzyme, combining a Superfamily 2 (SF2) helicase domain and an HD-type nuclease domain. After binding, Cas3 unwinds the target DNA in a 3′ to 5′ direction and progressively degrades the non-target strand, effectively shredding large regions of DNA downstream of the initial binding site [43,44].

This processive, unidirectional degradation makes Cas3 uniquely suited for large genomic deletions, often extending several kilobases. In contrast to the precise double-stranded cuts made by Cas9 or Cas12, Cas3 acts more like a molecular woodchipper, excising long stretches of foreign DNA—a feature critical for microbial immunity against invasive phages and plasmids [43,44].

From a biotechnology perspective, the complexity of Type I systems initially posed challenges for adaptation to genome editing in eukaryotic cells. However, recent innovations have enabled the successful reconstitution of the Cascade-Cas3 machinery in mammalian systems, allowing researchers to program broad deletions of non-coding DNA, repetitive sequences, and pathogenic genomic elements. For instance, engineered Type I systems have been used to remove pathogenic repeat expansions (e.g., in Huntington’s disease models) [45], perform synthetic genome minimization [46,47], and investigate topologically associating domain (TAD) boundaries in chromatin [48].

Recent advances in synthetic biology have begun to harness this powerful mechanism beyond simple cleavage. One emerging application involves fusing Cas3 to enzymatic modifiers, such as cytidine deaminases, to create tools for regional hypermutation. In one proof-of-concept system, Cas3 was engineered to carry a cytidine deaminase domain, enabling it to unwind and mutagenize extensive DNA segments, including entire metabolic operons. This approach achieved an approximately 350-fold enrichment of mutations within the targeted region compared to the background mutation rate with an enhanced average density of 0.3 mutations per kilobase—which is sufficient to evolve novel phenotypes or metabolic functions in a single round of selection [49].

Such fusion constructs have the potential to revolutionize in situ pathway engineering, adaptive evolution, and combinatorial mutagenesis, particularly in microorganisms and synthetic genomes. Moreover, the helicase function of Cas3 could be leveraged in the future for coupling with base editors, prime editing systems, or epigenome modifiers, where active DNA unwinding is a rate-limiting step. As delivery and specificity challenges are addressed, Cas3’s capabilities position it as a multi-modal genome engineering platform, extending the CRISPR toolkit into applications that demand high-throughput, long-range, and multiplexed editing.

#### 3.1.5. Cas10: A Central Integrator of RNA Sensing and DNA Defense

Cas10 is the catalytic core of type III CRISPR-Cas systems (Class 1), which are unique in their ability to target both RNA and DNA in a transcription-dependent manner. Unlike type I systems (e.g., Cas3) or type II systems (e.g., Cas9), Cas10 operates within a multi-subunit ribonucleoprotein complex, which is typically referred to as Csm (Type III-A) or Cmr (Type III-B). These complexes incorporate Cas10 along with multiple small Cas proteins (e.g., Cas5, Cas6, Csm3) and a guide crRNA, which together recognize and bind RNA targets that are transcribed from foreign DNA [49].

Once the crRNA binds to its complementary RNA transcript, Cas10 is allosterically activated and triggers two major immune functions:DNA Cleavage: Upon RNA recognition, Cas10 initiates degradation of the template DNA strand encoding the target RNA. This process requires active transcription and is tightly regulated to minimize autoimmunity. The DNA cleavage mechanism involves the HD nuclease domain of Cas10, which generates localized double-stranded breaks at the DNA locus only when the matching RNA is detected [49].Cyclic Oligoadenylate (cOA) Synthesis: The Palm domains of Cas10 catalyze the ATP-dependent synthesis of cyclic oligoadenylates (cOAs)—molecular second messengers structurally similar to cyclic AMP. These cOAs then bind and activate auxiliary CRISPR-associated Rossmann fold (CARF) domain nucleases, such as Csm6 or Can2, which degrade RNA nonspecifically in a powerful collateral response to eliminate phage transcripts or mobile genetic elements [49]. The “palm domain” refers to a specific structural region within the enzyme that is crucial for its catalytic activity. It is part of the larger “palm, fingers, and thumb” structure, resembling a right hand, that forms the core of the enzyme’s active site [50].

This two-tiered immune response—targeted DNA destruction and non-targeted RNA degradation—gives type III systems with Cas10 a fail-safe defense strategy, allowing them to dynamically adjust their activity based on ongoing transcriptional threats. This capability is particularly well suited for phage-host co-evolution, where the timing and regulation of nuclease activity can determine the success of bacterial immunity [51].

Recent bioengineering efforts have focused on reprogramming Cas10 and type III complexes for applications beyond microbial immunity, including the following:Synthetic biology circuits: The cOA signaling cascade has been harnessed to develop programmable biosensors, where specific RNA triggers can activate downstream effector enzymes, fluorescent readouts, or therapeutic payloads [52].Dual-mode regulators: Cas10’s coupling of RNA sensing with DNA cleavage has inspired systems that could conditionally regulate gene expression or edit DNA only in specific transcriptional states, improving safety and specificity in mammalian genome editing [53,54].Antiviral defense systems: Type III systems are being explored as programmable platforms for RNA virus detection and neutralization with the advantage of recognizing and degrading actively replicating viruses [55].

Moreover, comparative structural studies have revealed similarities between Cas10’s Palm domains and other nucleotidyl transferases, prompting interest in Cas10-based diagnostics analogous to Cas12 and Cas13 platforms [56].

In summary, Cas10 functions as a molecular switchboard, translating RNA recognition into both DNA destruction and global RNA silencing through second messenger signaling. Its versatility makes it a promising component for next-generation CRISPR tools that require transcriptional context-awareness, modular signaling, or multi-layered regulation.

#### 3.1.6. CasΦ: A Hypercompact CRISPR Effector from Bacteriophages for Therapeutic Delivery

CasΦ (pronounced “Cas-phi”), also known as Cas12j, is an ultra-compact Class 2 CRISPR-Cas effector discovered in the genomes of large bacteriophages (often called “megaphages”). At approximately 70 kDa; CasΦ is less than half the size of Cas9 (~160 kDa) and even smaller than Cas12a, making it the smallest known DNA-targeting CRISPR enzyme with programmable activity. This remarkable size reduction, coupled with its retained double-stranded DNA cleavage capability, has generated substantial interest in CasΦ as a next-generation genome editing tool—especially in contexts where delivery size constraints are paramount [56].

Despite its compact size, CasΦ maintains the essential molecular machinery required for RNA-guided DNA targeting and cleavage. It uses a single crRNA, like Cas12, to identify complementary DNA sequences adjacent to a PAM motif. Upon binding, CasΦ introduces double-stranded breaks (DSBs), although the precise nature of its cleavage pattern (blunt vs. staggered ends) is still under investigation. Its RuvC-like nuclease domain performs the catalytic cleavage, and structural studies suggest CasΦ maintains a minimal yet efficient architecture for target recognition and strand scission [56].

The small size of CasΦ is a critical advantage for therapeutic delivery platforms, particularly adeno-associated virus (AAV) vectors, which have a strict ~4.7 kb packaging limit. Cas9 often requires dual-vector systems or truncated promoters and regulatory elements, complicating delivery and reducing efficiency. In contrast, CasΦ allows for the inclusion of full-length guide RNAs and regulatory elements within a single vector, streamlining gene therapy development for in vivo applications [56].

While CasΦ demonstrates robust activity in vitro and in bacterial systems, its editing efficiency in mammalian cells remains an area of active optimization. Strategies to improve nuclear localization, enhance crRNA stability, and engineer more effective PAM recognition have begun to yield improvements. Additionally, efforts are underway to fuse CasΦ to base editing or transcriptional regulation domains, capitalizing on its small footprint to create multifunctional tools for precise genomic or epigenomic modulation [57,58].

Recent studies have also explored the potential of CasΦ for multiplex genome editing, owing to its ability to process CRISPR arrays and operate with minimal accessory components [23]. Its prokaryotic phage origin raises intriguing questions about horizontal gene transfer, evolutionary adaptation, and the potential of bacteriophage-encoded CRISPR systems as a largely untapped resource for tool development [59].

In summary, CasΦ represents a promising frontier in genome engineering, combining programmable specificity, dsDNA cleavage capability, and unprecedented compactness. As delivery remains one of the greatest challenges in clinical CRISPR applications, CasΦ offers a strategic solution for gene editing in difficult-to-target tissues, such as the central nervous system, eye, and liver, through viral or non-viral delivery vectors.

#### 3.1.7. CasΨ (CasPsi): A Dual-Targeting, Compact CRISPR Effector with Unique Versatility

CasΨ (pronounced Cas-Psi), also called Cas12j, is a recently characterized Class 2 CRISPR-Cas effector that exhibits an unusual and highly versatile activity profile—it can cleave both DNA and RNA, depending on the configuration of its guide RNA and surrounding biochemical environment. This flexible substrate specificity, along with its compact size (smaller than Cas9 and Cas12), positions CasΨ as a promising multifunctional genome engineering tool with broad potential in research, therapeutics, and diagnostics [59].

Unlike most Cas effectors that are highly specialized—such as Cas9 and Cas12 for DNA cleavage or Cas13 for RNA—CasΨ demonstrates conditional target preference, which is determined by factors such as the guide RNA structure, PAM or PFS (protospacer flanking site) sequence, and cofactor availability. Experimental studies suggest that subtle changes in the crRNA scaffold can toggle CasΨ’s activity between DNA and RNA cleavage modes. This makes it a programmable dual-mode nuclease that is adaptable for targeting diverse nucleic acid species within a single system [60].

CasΨ is phylogenetically distinct from previously described Cas families, indicating it likely evolved independently and may represent a new evolutionary branch within Class 2 CRISPR systems. This novelty opens the door to the further bioinformatic mining of microbial genomes for related effectors with similarly unexpected or hybrid functionalities [61]. Structurally, CasΨ is small and streamlined, making it ideal for in vivo delivery, including through AAV vectors, lipid nanoparticles, or exosomes—a major advantage for translational applications [61].

Emerging research indicates that CasΨ can be engineered or fused with functional domains, such as deaminases or transcriptional regulators, to serve as a platform for base editing, RNA editing, and CRISPRa/CRISPRi regulation. In diagnostic contexts, CasΨ’s programmable activity and collateral cleavage potential could enable dual DNA/RNA detection, making it particularly attractive for pathogen diagnostics, oncogene surveillance, or biomarker sensing in heterogeneous samples [61].

In summary, CasΨ stands out for its dual substrate flexibility, phylogenetic novelty, and compact format. As efforts to characterize and optimize its targeting rules continue, CasΨ is poised to become a next-generation CRISPR tool with applications across genome editing, transcriptome modulation, biosensing, and synthetic biology.

### 3.2. TIGR-Tas Mechanism and Structure

TIGR-Tas (Transposon-encoded Integrative Guide RNA-directed RuvC-like Transposase-associated Systems) represents a recently discovered and mechanistically distinct class of RNA-guided DNA nucleases. Unlike traditional CRISPR systems such as Cas9 and Cas12 that rely on single-guide RNAs and protospacer adjacent motif (PAM) recognition, TIGR-Tas utilizes a dual-guide RNA known as tigRNA. Each tigRNA molecule contains two target-recognition spacers of approximately 18 nucleotides, which are separated by a short linker loop. This dual-targeting architecture allows TIGR-Tas systems to recognize and bind both DNA strands independently of PAM sequences, thereby vastly expanding the potential targeting scope compared to conventional CRISPR effectors [2].

Upon binding to its target DNA, the TIGR-Tas effector protein—termed Tas—forms a symmetric dimer with each monomer engaging one of the guide regions. The protein uses RuvC-like nuclease domains to introduce a double-stranded break with defined 8-nucleotide 3′ overhangs in contrast to the blunt ends created by Cas9 or the variable overhangs generated by Cas12. This structural precision may offer advantages in ligation efficiency and in repair template design for precise genome editing. Notably, cryo-electron microscopy (cryo-EM) studies have revealed the presence of a Nop (nucleolar protein) domain within Tas, which is structurally homologous to the RNA-binding domain found in box C/D snoRNPs—eukaryotic complexes responsible for site-specific rRNA methylation. This unexpected structural similarity suggests that TIGR-Tas may have evolved convergently or shares an ancient evolutionary link with eukaryotic RNA-guided silencing machineries, highlighting the potential functional versatility of RNA-protein complexes across domains of life [2].

Functionally, the system has already demonstrated editing activity in human cells using a variant called TasR, achieving genome modification efficiencies between 0.8% and 3.6% without optimization. While modest, these results are comparable to early-generation CRISPR-Cas9 systems, underscoring the potential of TIGR-Tas as a next-generation genome editing platform. Additionally, the compact size of Tas proteins—typically less than half the size of Cas9—makes them particularly well suited for AAV-based delivery and other size-constrained delivery vehicles [2].

The TIGR-Tas system offers three major advantages over traditional CRISPR effectors. First, its PAM independence liberates it from one of the major design constraints that limits Cas9 and Cas12. Second, its compact size enhances its potential for in vivo applications, particularly in gene therapy. Third, its defined 3′ overhangs may improve precision in DNA ligation and homology-directed repair outcomes. These features collectively position TIGR-Tas as a powerful tool for both research and clinical applications [2].

In comparison, Cas9 requires PAM recognition (typically NGG) and generates blunt-ended double-stranded breaks guided by a single sgRNA. Cas12 requires a T-rich PAM and produces staggered cuts but still relies on PAM constraints. Cas13, although programmable, targets RNA rather than DNA, limiting its genome-editing capabilities. Cas3 is a processive helicase-nuclease that shreds DNA in a unidirectional manner, which is useful for large deletions but not precise edits. Cas10 functions in multi-subunit complexes with cyclic oligonucleotide signaling and lacks the simplicity of Class 2 effectors [2].

Other compact effectors such as Cas14 and CasΦ offer small size advantages, but each has limitations: Cas14 targets only single-stranded DNA and exhibits robust collateral activity, making it more suitable for diagnostics than editing. CasΦ, while extremely compact, still requires a PAM and is not yet optimized for high efficiency editing in mammalian cells. TIGR-Tas uniquely combines the desirable features of compact architecture, PAM independence, programmable dual-strand cleavage, and defined overhang generation, making it one of the most promising new additions to the CRISPR genome editing toolkit [2].

### 3.3. Evolutionary Implications

The TIGR-Tas system presents a striking departure from the classical CRISPR-Cas model of adaptive immunity, suggesting the existence of a distinct evolutionary pathway for RNA-guided nucleic acid targeting. Unlike canonical CRISPR systems, which clearly evolved in prokaryotes as a form of adaptive immune memory—capturing foreign DNA fragments as spacers in CRISPR arrays and mobilizing Cas effectors in response to re-infection—TIGR-Tas appears to have no associated CRISPR locus or spacer integration machinery, indicating that it did not originate from the same adaptive immunity framework [2].

Instead, mounting structural and functional evidence suggests that TIGR-Tas may have evolved from—or shares deep homology with—RNA-guided transcriptional silencing complexes, which is similar to eukaryotic box C/D small nucleolar ribonucleoproteins (snoRNPs). The C/D domains of snoRNAs have roles beyond RNA splicing, including protein binding, RNA interaction, and influencing gene expression [62]. The presence of a Nop domain in Tas proteins, which is commonly found in RNA-binding components of snoRNPs involved in rRNA methylation and pre-RNA processing, points to a shared evolutionary ancestry or convergent functional design [2]. This raises the possibility that RNA-protein complexes capable of sequence-directed cleavage or modification may have evolved multiple times independently potentially from a primordial RNA world where RNA played both catalytic and information roles [2].

The absence of PAM dependency in TIGR-Tas systems also suggests a functional divergence from CRISPR-based target discrimination mechanisms. PAM motifs in CRISPR systems serve as safeguards to prevent self-targeting, particularly in systems that integrate self-derived sequences as spacers. Because TIGR-Tas does not rely on such self/non-self-discrimination, its mechanism may be more ancient or fundamentally different, perhaps adapted to transient, non-heritable targeting such as transcriptional silencing or temporary DNA modification [2].

Intriguingly, TIGR-Tas effectors have been identified not only in bacteria and archaea but also in phages and prophages, where they may play roles in interspecies genetic warfare, viral competition, or anti-defense strategies. This suggests that horizontal gene transfer and viral innovation may have played a key role in the dissemination and functional diversification of TIGR-Tas elements. It is plausible that these systems evolved as programmable restriction-like nucleases to manipulate or degrade host genomes, interfere with host defense systems (including CRISPR-Cas), or counteract rival phages during co-infection [2].

The dual-spacer tigRNA architecture and dimeric symmetry of the Tas protein further support the idea that TIGR-Tas effectors may predate or parallel the modularity seen in eukaryotic RNAi systems. Their discovery broadens the conceptual framework for RNA-guided biology, showing that nature has independently evolved compact, programmable, RNA-guided nucleases multiple times in diverse biological contexts [2].

As more TIGR-Tas variants are discovered across phage and microbial genomes, comparative structural and phylogenetic studies will be critical to unravel their evolutionary trajectory. The system’s distinct origin, unusual structure, and programmable function place it at the intersection of prokaryotic immunity, RNA biology, and molecular evolution, potentially illuminating new paradigms in genome regulation and defense.

## 4. Future Directions

TIGR-Tas represents a promising next-generation genome editing platform with potential advantages in specificity, flexibility, and delivery. To realize this potential, several avenues require urgent exploration:Engineering for Efficiency: Protein engineering and tigRNA optimization may dramatically enhance Tas activity in mammalian systems. Creating catalytically enhanced or base-editing variants is a near-term goal.Targeting Fidelity: The dual-guide architecture of tigRNA may enhance target specificity by eliminating the need for a protospacer adjacent motif (PAM), which is required in Cas9-based systems to prevent self-targeting. In the TIGR-Tas system, self-recognition is avoided intrinsically by the paired-spacer design, removing the evolutionary pressure for PAM discrimination. Importantly, the absence of a PAM requirement theoretically enables TIGR-Tas to target any RNA sequence, offering greater flexibility than CRISPR systems, which are constrained to sequences flanked by specific PAM motifs, or divergent PAM motifs such as Nme2Cas9 with a dinucleotide (N_4_CC) PAM requirement [63]. While this design potentially reduces off-target effects and expands the editable transcriptome, rigorous validation—such as GUIDE-seq or related genome-wide off-target profiling—is still necessary to assess and compare the fidelity of TIGR-Tas with that of established CRISPR systems like Cas9 [64,65].Delivery Strategies: Given the compact nature of Tas effectors (~¼ Cas9), AAV, LNP, and minicircle delivery strategies are likely to be effective—potentially enabling in vivo editing with smaller payloads.Functional Expansion: Like dCas9 and Cas12a, Tas proteins may be modified for applications in epigenome editing, transcriptional modulation, or nucleic acid detection.Discovery of Related Systems: Metagenomic mining may reveal TIGR-Tas relatives with altered cleavage logic, RNA targets, or multi-effector synergy. Their presence in phages suggests a broader ecological role that may include counter-defense or mutualism.Cleavage of RNA:DNA hybrids: There is no report that the TIGR-Tas system can cleave RNA:DNA hybrids. However, TasR exhibited no detectable nuclease activity against single-stranded RNA (ssRNA) or double-stranded RNA (dsRNA) substrates even when these contained sequence matches to spacer A, spacer B, or both. In contrast, TasR efficiently and precisely cleaved single-stranded DNA (ssDNA) targets that harbored complementary sequences to either spacer A or spacer B, indicating a strong preference for DNA substrates over RNA and highlighting its specific recognition of ssDNA guided by dual-spacer tigRNAs [2].Improving target specificity: While the current TIGR-Tas system is limited to ~20 nucleotide recognition sequences per tigRNA, an important parallel can be drawn from CRISPR-Cas9 technology, where the development of a nickase version of Cas9 (Cas9n) enabled dual-guide strategies. In such approaches, two sgRNAs direct nickase Cas9 enzymes to adjacent sites on opposite DNA strands, leading to a double-strand break only when both guides are correctly positioned [66]. This strategy significantly improves specificity by requiring dual recognition events. Similarly, a nickase version of TasR has been characterized, which cleaves only one DNA strand. This opens the door to analogous dual-tigRNA designs for TIGR-Tas, where two tigRNAs—each guiding a nickase TasR to nearby sites—could be used to generate targeted cleavage only when both tigRNAs are present. This dual-nicking approach would effectively increase the total recognition sequence beyond 20 bases, thereby reducing the likelihood of off-target effects and enhancing overall specificity, just as has been demonstrated for Cas9. The further development of dual-nickase TIGR-Tas systems could allow the precise and highly specific targeting of genomic loci without the constraint of PAM motifs, offering a compelling alternative to existing CRISPR technologies.

The evolution of genome editing is entering a phase of diversification. TIGR-Tas may not replace CRISPR-Cas systems, but its distinct properties offer complementary and, in some applications, superior performance for precise, compact, and programmable DNA manipulation.

## Data Availability

No new data were created or analyzed in this study.

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
