# Peer review of "TIGR-Tas and the Expanding Universe of RNA-Guided Genome Editing Systems: A New Era Beyond CRISPR-Cas"

_genes, 2025, doi:10.3390/genes16080896_

Round 1

Reviewer 1 Report

Comments and Suggestions for Authors

This review is well written and it is a timely novel need for therapeutic tool.

--Please mention what are the additional details you have mentioned in this review apart from the previous reports

--Since the Tas proteins contain box C/D small nucleolar ribonucleoproteins (snoRNPs) domain, please discuss does this affect splicing?

--Because there is no requirement of PAM, can this system cleave any part of the DNA? Repeat regions?

--If there are previous reports available, please mention what are the studies needed to be done for the efficiency of Tas protein.

--Since the recognition is limited to 20 bases with the current TIGR-Tas system, are there any  possibilities for larger recognition sequence? Will this help to minimize off target effect compared to CRISPR Cas9 system.

--Please provide what is the length of the scaffold/Spacer sequence needed for single guide RNA needed for activity. This will help for synthesis of single guide RNA.

--Are there any evidence that will the TIGR-Tas system cleave DNA-RNA Hybrid complexes?

--If there is any study comparing the efficiency of Cas9 cleavage vs Tas cleavage, please mention it.

--Briefly describe the nme2Cas9 system also in this review.

Author Response

TIGR Tas review responses

July 21, 2025

Reviewer 1:

Comment 1: This review is well written and it is a timely novel need for therapeutic tool.

--Please mention what are the additional details you have mentioned in this review apart from the previous reports

Response 1: There is only one peer reviewed paper on TIGR-Tas and I only included details from that reference [1]. There are several non-peer reviewed reports, such as in MIT Press, which I did not cite since they are not peer reviewed. I did not include any details from the non-peer reviewed journals or from online seminars from the senior author.

Comment 2: --Since the Tas proteins contain box C/D small nucleolar ribonucleoproteins (snoRNPs) domain, please discuss does this affect splicing?

Response 2: The C/D domains of snoRNAs have roles beyond RNA splicing, including protein binding, RNA interaction, and influencing gene expression. This has been added to the text on page 15 with a new reference [2].

Comment 3: --Because there is no requirement of PAM, can this system cleave any part of the DNA? Repeat regions?

Response 3: I added this to the Future Directions:

  1. Targeting Fidelity: Targeting Fidelity: The dual-guide architecture of tigRNA may enhance target specificity by eliminating the need for a protospacer adjacent motif (PAM), which is required in Cas9-based systems to prevent self-targeting. In the TIGR-Tas system, self-recognition is avoided intrinsically by the paired-spacer design, removing the evolutionary pressure for PAM discrimination. Importantly, the absence of a PAM requirement theoretically enables TIGR-Tas to target any RNA sequence, offering greater flexibility than CRISPR systems, which are constrained to sequences flanked by specific PAM motifs, or divergent PAM motifs such as Nme2Cas9 with a dinucleotide (N4CC) PAM requirement [3]. While this design potentially reduces off-target effects and expands the editable transcriptome, rigorous validation—such as GUIDE-seq or related genome-wide off-target profiling—is still necessary to assess and compare the fidelity of TIGR-Tas with that of established CRISPR systems like Cas9 [4, 5].

Comment 4: --If there are previous reports available, please mention what are the studies needed to be done for the efficiency of Tas protein.

Response 4: This is addressed above in Comment 3 response.

Comment 5: --Since the recognition is limited to 20 bases with the current TIGR-Tas system, are there any  possibilities for larger recognition sequence? Will this help to minimize off target effect compared to CRISPR Cas9 system.

Response 5: I added this to future directions:

Improving target specificity: While the current TIGR-Tas system is limited to ~20 nucleotide recognition sequences per tigRNA, an important parallel can be drawn from CRISPR-Cas9 technology, where the development of a nickase version of Cas9 (Cas9n) enabled dual-guide strategies. In such approaches, two sgRNAs direct nickase Cas9 enzymes to adjacent sites on opposite DNA strands, leading to a double-strand break only when both guides are correctly positioned [6]. This strategy significantly improves specificity by requiring dual recognition events. Similarly, a nickase version of TasR has been characterized, which cleaves only one DNA strand. This opens the door to analogous dual-tigRNA designs for TIGR-Tas, where two tigRNAs—each guiding a nickase TasR to nearby sites—could be used to generate targeted cleavage only when both tigRNAs are present. This dual-nicking approach would effectively increase the total recognition sequence beyond 20 bases, thereby reducing the likelihood of off-target effects and enhancing overall specificity, just as has been demonstrated for Cas9. Further development of dual-nickase TIGR-Tas systems could allow precise and highly specific targeting of genomic loci without the constraint of PAM motifs, offering a compelling alternative to existing CRISPR technologies.

Comment 6: --Please provide what is the length of the scaffold/Spacer sequence needed for single guide RNA needed for activity. This will help for synthesis of single guide RNA.

Response 6: I added this information to the Graphic Abstract: spacer A and B (9-12 nt), edge repeat/loop repeat (8-12 nt).

Comment 7: --Are there any evidence that will the TIGR-Tas system cleave DNA-RNA Hybrid complexes?

Response 7: I added this to the future studies section:

Cleavage of RNA:DNA hybrids: There is no report that the TIGR-Tas system can cleave RNA:DNA hybrids. However, TasR exhibited no detectable nuclease activity against single-stranded RNA (ssRNA) or double-stranded RNA (dsRNA) substrates, even when these contained sequence matches to spacer A, spacer B, or both. In contrast, TasR efficiently and precisely cleaved single-stranded DNA (ssDNA) targets that harbored complementary sequences to either spacer A or spacer B, indicating a strong preference for DNA substrates over RNA and highlighting its specific recognition of ssDNA guided by dual-spacer tigRNAs [1].

Comment 8: --If there is any study comparing the efficiency of Cas9 cleavage vs Tas cleavage, please mention it.

Response 8: This was discussed above as an needed experiment.

Comment 9: --Briefly describe the nme2Cas9 system also in this review.           

Response 9: I added nme2Cas9 to response 2.

Reviewer 2:

Comment 1: This is an excellent, highly informative review of a complex literature. I have only a few rather trivial questions and suggestions for improvement.

1.Line 162 mentions “HNH nuclease domains (His-Asp-His)” but I think it should be His-Asn-His, since N refers to asparagine, not aspartic acid.

Response 1: Good catch! I corrected this.

Comment 2: Line 301 mentions the density of 0.3 mutations per kilobase, but it was not clear whether this rate is the background mutation rate or the enriched mutation rate, which are being compared in this discussion.

Response 2: This is the enhanced rate. I changed the sentence as following:

This approach achieved an approximately 350-fold enrichment of mutations within the targeted region compared to the background mutation rate, with an enhanced average density of 0.3 mutations per kilobase—sufficient to evolve novel phenotypes or metabolic functions in a single round of selection [7].

Comment 3: 3.The phrase “CRISPR arrays” come up several times in the review. In order to make the review as helpful as possible for the uninitiated reader, a sentence or two explaining what these are could be added.

Response 3: I added “Crispr arrays” and “Tigr arrays” to the graphic abstract. This should clarify what they are.

New references:

  1. Faure, G., et al., TIGR-Tas: A family of modular RNA-guided DNA-targeting systems in prokaryotes and their viruses. Science, 2025. 388(6746): p. eadv9789.
  2. Cheng, Y., et al., A non-canonical role for a small nucleolar RNA in ribosome biogenesis and senescence. Cell, 2024. 187(17): p. 4770-4789 e23.
  3. Edraki, A., et al., A Compact, High-Accuracy Cas9 with a Dinucleotide PAM for In Vivo Genome Editing. Mol Cell, 2019. 73(4): p. 714-726 e4.
  4. Lazzarotto, C.R., et al., Population-scale cellular GUIDE-seq-2 and biochemical CHANGE-seq-R profiles reveal human genetic variation frequently affects Cas9 off-target activity. bioRxiv, 2025.
  5. Tsai, S.Q., et al., GUIDE-seq enables genome-wide profiling of off-target cleavage by CRISPR-Cas nucleases. Nat Biotechnol, 2015. 33(2): p. 187-197.
  6. Ran, F.A., et al., Double nicking by RNA-guided CRISPR Cas9 for enhanced genome editing specificity. Cell, 2013. 154(6): p. 1380-9.
  7. Zimmermann, A., et al., A Cas3-base editing tool for targetable in vivo mutagenesis. Nat Commun, 2023. 14(1): p. 3389.

Reviewer 2 Report

Comments and Suggestions for Authors

This is an excellent, highly informative review of a complex literature. I have only a few rather trivial questions and suggestions for improvement.

1.Line 162 mentions “HNH nuclease domains (His-Asp-His)” but I think it should be His-Asn-His, since N refers to asparagine, not aspartic acid.

2.Line 301 mentions the density of 0.3 mutations per kilobase, but it was not clear whether this rate is the background mutation rate or the enriched mutation rate, which are being compared in this discussion.

3.The phrase “CRISPR arrays” come up several times in the review. In order to make the review as helpful as possible for the uninitiated reader, a sentence or two explaining what these are could be added.

Author Response

(The authors gave the same response as above.)

Round 2

Reviewer 1 Report

Comments and Suggestions for Authors

Comments are prompt.